# Modeling Pulsed High-Power Spikes in Tunable HV Capacitive Drivers of Piezoelectric Wideband Transducers to Improve Dynamic Range and SNR for Ultrasonic Imaging and NDE

**DOI:** 10.3390/s21217178

**Published:** 2021-10-28

**Authors:** Antonio Ramos, Abelardo Ruiz, Enrique Riera

**Affiliations:** R&D Group “Ultrasonic Systems and Technologies”, Institute ITEFI (CSIC), Serrano 144, 28006 Madrid, Spain; artuscu@hotmail.com (A.R.); enrique.riera@csic.es (E.R.)

**Keywords:** efficient ultrasonic transceivers, wideband piezoelectric transducers, industrial NDE, medical imaging, pulsed high-power spikes, HV capacitive-discharge drivers, high-current driving, net dynamic range, NDRA

## Abstract

The signal-to-noise ratios (SNR) of ultrasonic imaging and non-destructive evaluation (NDE) applications can be greatly improved by driving each piezoelectric transducer (single or in array) with tuned HV capacitive-discharge drivers. These can deliver spikes with kW pulsed power at PRF ≈ 5000 spikes/s, achieving levels higher even than in CW high-power ultrasound: up to 5 kW_pp_. These conclusions are reached here by applying a new strategy proposed for the accurate modeling of own-design re-configurable HV capacitive drivers. To obtain such rigorous spike modeling, the real effects of very high levels of pulsed intensities (3–10 A) and voltages (300–700 V) were computed. Unexpected phenomena were found: intense brief pulses of driving power and probe emitted force, as well as nonlinearities in semiconductors, though their catalog data include only linear ranges. Fortunately, our piezoelectric and circuital devices working in such an intense regime have not shown serious heating problems, since the finally consumed “average” power is rather small. Intensity, power, and voltage, driving wideband transducers from our capacitive drivers, are researched here in order to drastically improve (∆ >> 40 dB) their ultrasonic “net dynamic range available” (NDRA), achieving emitted forces > 240 Newtons_pp_ and receiving ultrasonic signals of up to 76–205 V_pp_. These measurements of ultrasonic pulsed voltages, received in NDE and Imaging, are approximately 10,000 larger than those usual today. Thus, NDRA ranges were optimized for three laboratory capacitive drivers (with six commercial transducers), which were successfully applied in the aircraft industry for imaging landing flaps in Boeing wings, despite suffering acoustic losses > 120 dB.

## 1. Introduction

Each transducer channel (emitter and sensor) involved in ultrasonic imaging applications using piezoelectric transducers is usually classified as “low power”, e.g., in the case of the systems used for industrial NDE and medical diagnosis (echography) applications. This power level designation is adopted since the total integrated power consumed (from the HV electrical supply, V_0_) by each ultrasonic channel is of the order of the Watt, when it is integrated in its averaged value during 1 s (≈5000 driving spikes).

As a result, linear approaches are usually employed during the design and development of transducers and their electrical matching with HV drivers, for ultrasonic imaging [1,2,3,4]. However, in new highly efficient ultrasonic detection and imaging systems, where high-resolution (HR) and high SNR are required, a capacitive-discharge HV scheme inductively tuned [5,6] must be considered. This is the best way to obtain spikes with pulsed high-power, e.g., 1000–5000 W_pp_/spike and 350–700 V, at the usual pulse repetition frequency (PRF). These high driving values are needed, owing to the low transducer motional resistance R_m_ (tens/hundreds Ω) when radiating in the more usual tested media, and also its rather high input capacitance (nF).

Those high-power values are achieved here with a limited size in driving electronics. For a typical case, e.g., R_m_ = 150 Ω and V_0_ = 400 V, the energy stored in a 10 nF capacitor (0.8 milli-Joule), during 100 µs (with maximum charging current < 45 mA from HV supply, and this only in the firsts µs), can be used for a sudden pulsed high-power wideband driving. This will assure currents through the transducers of several A/spike, during its more usual widths, 100–500 ns for the MHz range. However, the final power consumed is only 800 µWatt/spike. In fact, an efficient driving with pulsed high power, for the high-NDRA applications mentioned in the abstract, will satisfy strong pulsed electrical demands across transducer terminals during all the repetitive driving times.

As an example, the supply of high pulsed currents, from HV drivers, must attain up to 10 A during some hundreds of ns. These extreme parametric values are not normally taken into account nowadays in the design and development of new efficient broadband ultrasonic emitters. It is very useful to include the effect of these parametric ranges during spike generation for achieving accurate modeling and simulation steps prior to the design of new ultrasonic imaging and NDE systems needing high SNR, i.e., high dynamic ranges.


*About the needed performances for an efficient driving of piezoelectric broadband transducers*


In order to obtain a quantitative determination of these extreme requirements, a rigorous analysis was made of some pulsed performances in our capacitive multi-drivers to be used in very efficient ultrasonic inspections. In future designs for new HR industrial and medical imaging applications, these high values will only appear during very short times in each one of the repetitive cycles for the needed transducer’s driving regime. A selected inductive parallel tuning and associated diode network (used here for approaching second-edge verticality in HV spikes to the first edge slope [5,6]) were also included.

Additionally, the peak voltages for driving in emission should be rather high, to be able to guarantee amplitudes larger than the usual values in the received ultrasonic signals (tens/hundreds of mV). This will assure a resulting very high performance. Finally, the design of whole ultrasonic transceivers must seek to improve global SNR, as well as to achieve echo-pulses with a greater dynamic range than current ones, in order *to obtain clearer measures and images*. In new, more efficient applications for ultrasonic imaging, the total *average power* delivered to each PZT ultrasonic transducer or array element, from HV capacitive pulse generators, *can remain at a moderated value*. This would be in the order of a few Watts/transducer (for usual PRF), by maintaining the HV supply voltage ≤ 500 V.

Despite these rather limited values in driving voltage and final consumed power, very high-power phenomena of rather brief duration (hundreds of ns) can be observed by our transient analysis under the pulsed regime required for ultrasonic testing. Therefore, the pulsed electrical power, repetitively delivered (by means of short spikes) to each transducer, from an efficient capacitive driver (at conventional PRF: 2–5 kHz), will attain levels similar to those in single transducer of CW (continuous wave) high-power ultrasonic cases. For example, levels of up to 5 kW are reached during each short emission process in channels of very efficient pulsed NDE ultrasonic applications using PZT transducers.

As a result of the mentioned high values, intense transient effects will occur thousands of times/s, in new very efficient emission systems for HR images forming. In particular, high-peak electrical currents will occur through electronic drivers needed in those systems, together with elevated peaks of driving power and emitted force in the transducers. Additionally, temporary high peaks in other involved parameters, and notable short-duration non-linear electrical/piezoelectric responses, could appear due to the associated very intense pulsed working regime needed in these efficient HR imaging applications.

For example, due to these elevated parametric values and variation ranges registered in our efficient HV electronic drivers under study, some semiconductor devices in these NDE drivers can reach the saturation state under pulsed intensities of up to 5–10 Amperes. Therefore, these high levels could cause brief non-linear behaviors very distinct to those observed with low electrical currents. Moreover, these high currents and voltages (300–700 V), supported by some components, could lead to unexpected brief electronic responses. Thus, these huge parametric ranges were considered here in modeling and simulation tasks needed in the design processes of our new equipment for very efficient ultrasonic detection and imaging purposes, but with HV supply consumption < 4 W/channel.

These non (ideal/linear) behaviors were not expected in principle in a broadband ultrasonic channel using low-power transducers, since the total amount of consumed averaged power is rather small in these application types. However, the pulsed ultrasonic energy repetitively emitted by the transducers, and the instantaneous electrical power IEP (in each pulsed shot) delivered by each electronic HV driver (during repetitive very short emission time-periods), can attain levels even higher than those registered in typical CW high-power ultrasonic applications. For example, several kW of pulsed power delivered to each emitter transducer were observed in our laboratory for each individual driving.

For comparative purposes, in high-power (HP) ultrasonic applications [7] it is usual to register: (a) for the CW medium-power cases, driving peak values of 600–700 V in voltage, and 0.6–0.7 A in current; i.e., 400–500 W, as maximum, in power; (b) in very high-power ultrasonic equipment, values such as 1400–1800 V in voltage, 1–1.2 A in current, and 1.5–2 kW in power can be attained across terminals of the working ultrasonic transducers. These HP excitation values are smaller *than those needed in our*
*Pulsed High-Power PZT Emitters* (PHPPE), *which will be investigated in depth in this paper*.

A justification must be made in order to understand: (1) the causes of the pulsed *high-power*
*levels*
*required here*
*for driving*; and (2) the need to *design special driver schemes based on capacitive discharges*. The main reason for both requirements arises from the high electrical current levels that must be supplied from the HV drivers, *due to the rather low values of*
*the*
*input resistive impedances in* the wideband piezoelectric *transducers needed here*. In fact, they range from ≈50 to 300 Ω for ultrasonic radiations in the more usual media. Furthermore, the capacitances across the PZT transducer terminals are rather elevated, of the nF order.

All these data were measured across the terminals of the transducers used here (which were *inductively tuned in* their *series resonances*). This makes necessary the use of HV capacitive drivers. Other alternative non-capacitive options, only used to drive transducers with high input resistance, are not valid to be applied here due to the low resistances of load specified above, which demand high pulsed powers and currents.

To summarize all the above described: Some unexpected very high values in the pulsed regime (maximum electrical levels to be supported by driving electronic components and piezoelectric transducers) were taken into account for our rigorous analysis of the whole emission piezoelectric process. This must be achieved using the specific data of pulsed powers and electrical currents needed to efficiently drive ultrasonic detection and imaging cases, for medical and industrial applications with high resolution and SNR [8].


*Originality and main contributions of this paper with respect to the previous state of the art:*
▪It is shown for the *first time* that each *low-power piezoelectric emitter* (HV driver + PZT transducer), needed *for very efficient* ultrasonic imaging and NDE applications, *must work with very high power* and *current pulses* in some inner driver circuits: *up to 5 Kw and 10 A,* respectively.▪A *new strategy is proposed here for an accurate modeling* of the “pulsed high-power PZT emitters” (PHPPE), employing our *analytical and circuital models* created *to optimize those for a best fitting to* two frequency ranges, distinct tuning effects, and non-idealities in electrical and piezoelectric devices.▪Clearly justified here, for the *first time*, are *the* very notable *influences of motional origin* from a piezoelectric transducer, superimposed *on its originator HV driving spike*, with an *excellent agreement* registered between our *computer*
*modeling* and the resulting *experimental waveforms.*▪Our calculated and experimental *results show* a *dramatic widening in dynamic range* by *fitting our capacitive PHPPE, from* the results of the *modeling strategy we propose here*. Final *received* T–T *signals of huge amplitudes* (76 V_pp_ through a plastic piece, and 130–205 V_pp_ by direct transducers contact), and a *sharp HR image* in the NDE of a Boeing-777 wing flap (with >100 dB *net dynamic range*), are achieved. ▪These huge received amplitudes in voltage must be compared with those normally acquired in classical pulse-echo imaging and T–T NDE cases using *other PHPPE*, which usually range below 1 V_pp_ or even 100 mV_pp_. These *final ultrasonic levels are <<46 dB, with respect to our improved results*, to be shown here.▪A *new index is defined* here, the ultrasonic “net dynamic range available” (NDRA), which shows that *resulting net dynamic ranges can be*
*multiplied by (>10^4^)* from their usual values, using our modeling strategy, thus *optimizing the received signals* for three laboratory capacitive driver settings with six commercial transducers.
Summary of the next article sections

In Section 2.1, our schemes, designed for HR ultrasonic imaging and for the capacitive circuits to obtain their multi-channel high-current driving, are described. The motional piezoelectric influence of transducer vibrations on a measured −290 V spike is shown and analyzed. In Section 2.2, specific methods are presented for experiments and mathematical/circuital modeling and simulations of PHPPE PZT emitters. In Section 3: (a) High-power driving (at 400 V) and the resulting emitted mechanical results are shown, being comparable in magnitude to CW high-power cases. (b) Pulsed signals received in transmission (T–T) mode are investigated in transceivers with non-linear components. All our results, using tuned HV capacitive drivers of PHPPE, are discussed in Section 4.

## 2. Systems, Instrumental Methods, and Technologies Employed in the Paper

### 2.1. Electronic Systems and Technologies

We have integrated flexible electronic systems designed in our labs of CSIC to generate ultrasonic signals and synthesize the final images [5,8,9]. These systems can be used in distinct applications such as: (a) quality control of industrial pieces and materials [10], and (b) high-resolution (HR) ultrasonic diagnosis in structures and hospitals [8]. A general block diagram of their integration is shown in Figure 1.

This complex equipment, designed to improve ultrasonic resolutions, includes piezoelectric transducers and many electronic channels for emitting (drivers) very short HV spikes (≤500 ns) and receiving ultrasonic signals. Both E/R pulsed waveforms can be electronically focused and scanned, firstly in emission and later in reception stages, using multiple precision delayers and units for Mux-DMux, of the electrical HV spikes and of the corresponding ultrasonic echo-traces, as well. The channels involved can be dynamically controlled [5,8] during each successive driving and signal acquisition cycle (around 5000 times per second). Hence, these units can create multiple imaging lines, in a parallel way.

In each individual driver, a *selective* damping block works in cascade with a basic HV pulse generation circuit. This damping block notably reduces certain parasitic oscillations induced in the driving pulses, due to the more efficient emission matching circuits (for medical and NDE imaging applications) that include an inductive tuning [8,9,11]. In the case of pulse-echo mode, each parallel channel in Figure 1, for both E/R piezoelectric sections, can share a unique matching circuit (selective damping and tuning), due to only one of the two sections simultaneously working, either as an emitter or as a receiver.

Logically, the periods with high values in the delivered pulsed power are those related to each repetitive HV driving in all the channels involved. The HV excitation pulse amplitudes normally range from 300 to 500 V, but they can attain up to 700 V. 

The pulse repetition frequencies (PRF) for pulse-echo imaging vary between 2000 and 10,000 spikes/s, depending on the final imaging resolution (number of emitted foci and of scanning lines). The duration of each high-voltage driving spike varies from 100 to 500 ns in the more usual inspection cases.

These electrical characteristics assure an efficient transducer excitation repeated several thousands of times each second for all the channels, in order to obtain the elevated number of echo-A-scans that make it possible to perform the fast processes employed nowadays for the generation, electronic focusing and scanning, and fast processing of all the resulting pulsed ultrasonic beams involved in forming a HR ultrasonic image [8].

Each basic pulser typically includes a number of non-linear circuits, e.g., semiconductor devices [5,12,13]. These circuits have to support short pulsed intensities of up to 10 Amperes, and as a result, they can suffer high-saturation states, with possible non-linear responses. In fact, the characteristic curves of many electronic devices (used here) contemplate only linear responses under moderate intensity values (e.g., <1 Ampere). 

Photos of reconfigurable systems and electronic cards (such as those schematized in Figure 1, designed in our R&D CSIC group, to investigate transducer multi-driving and electronic focusing/scanning of ultrasonic beams) are depicted in Figure 2.

Fortunately, the piezoelectric devices, working in such a high-power transient regime, do not show serious heating problems (which could be potentially destructive), since the associated mean averaged power dissipation remains in the order of the Watt.

However, very high levels in both pulsed electrical intensities (2–10 A) and voltages in the HV output stages (300–700 V) could lead to transitory non-linear behaviors. These non-ideal effects will be considered in the working transient regime here used for achieving a precise modeling of the time and frequency responses in each HV driver of the *n* PZT channels, involved in these ultrasonic applications [12,14].

#### Our Circuits for High-Intensity Driving from HV Capacitive Discharges

The capacitive drivers designed for efficient ultrasonic emission include high-voltage negative ramp generators, proposed firstly through a thyristor SCR in [16] and after by us through a MOS–FET power transistor (MFT) connected to a “selective” damping network [5]. In this improved second case, very low output impedances and very short fall times (≈ 10 ns) can be reached. The ramp generator is connected in cascade with specific circuital networks [5] in order to achieve optimal electrical matching with wideband emitter transducers, in this way being able to reach high-power spikes to supply a very high current in efficient ultrasonic emission processes based on PZT devices. 

Figure 3 shows the blocks diagram of our efficient tuned driving strategy employed for optimizing broadband transducer responses in a frequency range up to 30 MHz, which permits the generation of intense narrow spikes of pulsed high power [8].

The specific circuital topologies we have designed for the HV ramp generator, pulse shaper, and “selective” (inductive) damping (shown in Figure 3) are detailed in Figure 4.

The HV waveforms across the terminals of transducer Z_T_ adopt the shape of a “spike” (shown in Figure 4b). A negative HV pulse with a rather vertical first edge (few nanoseconds) is followed by another positive edge with controllable short duration. 

The inductance L_0_ (and in some occasions, a Zener diode Z) are added [5] since they can improve the transit times of the final driving pulse across transducer terminals (Z_T_). The diode network D_1_–D_2_ eliminates possible high-level oscillations that would be originated by the (C–L_0_) resonance in the case of short duration spikes, after the launching of the main HV spike pulse. R_C_ can be in the order of tens of kΩ.

Finally, the resistance R_L_ of very low value (≈1 Ω) is added as an electrical protection under the high electrical currents registered through the output channel during driving.

It must be noted that this capacitive circuital configuration offers the possibility of generating pulses of HV (up to 500–700 V) and high pulsed power (order of k-Watt) across piezoelectric transducers in tuned series resonance conditions, even those having very low input resistive impedance (tens of ohms) and high capacitance in parallel (some nF).

It is usual, for ultrasonic imaging, to choose widths of spike from 100 to 500 ns, depending on the effective central working frequency of each specific transducer, which coincides with the maximum electrical conductance band of the piezoelectric device to be driven, in this current case with a very low driver output impedance.

### 2.2. Other Equipment and Transducer Devices Used for This Experimental Research

In addition to our own design systems for HV multi-driving, electronic focusing and scanning detailed in Figure 2, Figure 3 and Figure 4, the other instruments and devices employed in this work to perform the distinct experiments are the following:-For programing different settings in an HV driver and generating the needed spikes: A flexible laboratory ultrasonic HV transceiver (of our own design), with selectable electrical and electronic parametric setting, was configured for each experiment included in this work. This transceiver is shown in Figure 5a.-For measuring the frequency behavior of the complex input impedances in transducer terminals, a precision digital spectral analyzer of impedances Agilent 4294A (Santa Clara, CA, USA) was used, with an ample frequency range, from 40 Hz to 110 MHz, and a measurement accuracy of ±20 ppm for all the impedance range.-For displaying the E/R pulses (HF/HV spikes and pulsed ultrasonic signals) and to make their FFT transforms: A DSA 602A (Tektronix Inc., Beaverton, OR, USA) digitizing signal analyzer was used, with a flexible windowing. Its sampling rate was up to 2 GHz and the channel input impedance was the parallel of 10 MΩ//11 pF.-The wideband multilayer piezoelectric probes (mechanically backed and matched) used in experiments, were four TecalSA (Madrid, Spain) transducers designed specifically in our SSTU laboratory of CSIC, and four commercial NDE transducers (two Krautkrämer and two Panametrics). Photos of all of them are shown in Figure 5b.

The experiments carried out in our laboratory to generate the distinct received signals analyzed in this paper use dispositions based on the through-transmission (T–T) mode, by direct coupling of two coaxial transducers with the parallel faces of a plastic (PMMA) piece or using a double water jet coupling. This last option was employed for the industrial inspection of landing flaps in the wings of Boeing 777 airplanes. 

For the industrial experiments considered here, multi-transceivers of our HV tuned capacitive type were developed in CSIC by our group GSTU (IA), together with a processing and display unit by IA and IAI [17], which were industrially installed by the company Tecal SA in some testing plants of EADS-Boeing and Airbus-Eurocopter companies.

In Section 3 of this paper (*measured and simulated pulses of improved dynamic range**,*
*by capacitive driving*), other specific aspects will be detailed on the instrumental methods, where our tuned HV driver was successfully applied for industrial robotic NDE of landing flaps, overcoming very large total acoustic attenuation losses near 150 dB.

### 2.3. Our Methods for Theoretical Transient Analysis, Circuital Modeling, and Computational Simulation of Tuned HV Capacitive Drivers and Multilayer Piezoelectric Transducers

The first subject described here refers to our expressions in time and spectral domains about the driving voltage DV generated from inductively tuned HV capacitive driving. By means of these expressions, a linear analysis was carried out on the responses of this type of HV drivers, using classical methods. 

However, the presence of non-linear behaviors of electrical elements, in the circuital topologies analyzed in this work, prevents in many cases obtaining driving waveforms such as those encountered in the practice with our real equipment (shown in Figure 4 and Figure 5a) containing the above described electronic HV capacitive driver.


*Our Time and Spectral Analytical Expressions for HV Spikes created by Capacitive Drivers*


Assuming a linear behavior in the electronic emission stage of the ultrasonic imaging or NDE channels, the spike for driving voltage (DV) created by a capacitive driver can be analytically expressed by means of a time convolution between an exponential function (created, in practice, by a ramp generator as in Figure 3 and Figure 4) and the impulsive response IR_CN_ of the coupling network of the driver with the transducer:Driving Voltage (t) = IR_CN_ (t) ∗ V_0_ e^−t/τ^(1)
where V_0_ and τ are the amplitude and fall-time, respectively, of a ramp function which has a real time evolution near an exponential curve with a short fall time (≅10–15 ns).

The ultrasonic pulsed signals (which will be finally received) can be considered as a multiple convolution of the driving voltage appearing in (1) with the impulse responses (IR) of successive emitting and receiving piezoelectric processes in E/R transducer and the IR encountered by the signals through E/R aperture diffractions and propagation media.

There is an option, based on a mathematical description, for obtaining a whole analytical expression for practical driving cases, based on a circuital assumption applicable only in certain situations. However, in the other cases, that expression will continue being valid partially, but uniquely for the main part of DV, just the first negative HV pulse [8,9]. 

Despite this limitation, a rather simple possibility to model a driving waveform is to make a frequency domain analysis, under the assumption that the effects of both, R_L_ resistance (few ohms) and the diode network D_1_–D_2_ in Figure 4, can be disregarded, under certain conditions. Then, a time response can be obtained [8] using the inverse Laplace transform of our following analytical expression for DV(s):(2)$$\mathrm{DV}\left(s\right)\;=\;-{\mathrm{CV}}_{o}{\left[{\tau C}_{S}s^{2}+\;\left(C_{s}+\tau /R_{\mathrm{eq}}\right)s+\;\left(1/R_{\mathrm{eq}}+\tau /L_{0}\right)+\;\left(1/L_{0}\right)s^{-1}\right]}^{-1}$$
where C_S_ = C + C^0^_XT_, C^0^_XT_ being the transducer static capacitance, and R_eq_ is given by:(3)$$R_{\mathrm{eq}}=R_{\mathrm{XT}}{\;R}_{D}{\;R}_{\mathrm{SR}}/\left(R_{D}{\;R}_{\mathrm{SR}}+R_{\mathrm{XT}}{\;R}_{\mathrm{SR}}+R_{\mathrm{XT}}{\;R}_{D}\right)$$

In (3), R_eq_ is the electrical equivalent parallel of: (i) the electrical damping, R_D_, of the HV driver; (ii) the input resistance (R_SR_) of possible devices for display or amplification connected across transducer terminals; and (iii) the transducer resistive component in the motional branch (R_XT_), measured in its input terminals at frequency ω_s_ (series resonance).
R_XT_ = 1/max G(ω) = [G(ω_s_)]^−1^(4)

From the inverse transform IFFT of the expression (2), for s = jω, a rather oscillatory waveform is obtained in time domain for DV(t); (see Figure 8 in [9]). 

The problem with this limited option is that, for short spikes, the oscillations do not appear exponentially damped quickly after the first half-cycle, as happens in our real circuit (after the big pulse in DV), due to the complex filtering effects originated from the diode network D_1_–D_2_, which have not been considered in this analytical approach. 

Nevertheless, at this point of the analysis, these real effects could be simply approximated, emulating the cancellation of the oscillatory response, by setting it to zero from the end of the first ascendant spike edge, just after launching the high-voltage negative spike, and thus preventing an undesired lengthening on the driving pulse [9].

However, this “corrected” option is still somewhat inaccurate in respect to our experimental waveforms measured in the laboratory for the “tuned” HV capacitive drivers. Thus, a more accurate numerical simulation of this complex HV capacitive driving would be required for a better quantization of the pulsed electrical current and power through transducers. To attain this aim, we found other non-analytical options, based on circuital models with non-linear semiconductors under an electrical high-current regime.


*Circuital solutions applicable for non-linear modeling and numerical simulations of tuned capacitive generators driving piezoelectric transducers with pulsed high-power spikes*


An accurate solution for non-linear modeling of tuned HV capacitive drivers is to apply a software developed by the paper authors, to implement P-Spice circuital simulations of all involved high-frequency (electronic/piezoelectric/ultrasonic) systems.

In Figure 6, our first scheme is shown, including several non-ideal behaviors, which is very useful to improve the modeling of the HV driving of piezoelectric transducers in the high frequency (HF) range (from 3 to 30 MHz). 

In this scheme, the parasitic inductances (L1, L2, L3) related to specific printed circuit tracks are impedances that must be taken into account due to their possible HF attenuation effects. These parasitic inductances range usually in the order of 50–200 µH. Inductances L (4–5) also include the cable connecting the transducer with the HV spike driver, and they could be substituted by a transmission line if a coaxial cable were used for the connection.

The devices Di (i = 1…6) are signal diodes that prevent the formation of harmful oscillatory signals, originating mainly from the series resonance between *C*d and Lp, and that (as a first effect) allow only the first half-cycle (just, the negative spike) to pass for performing the HV transducer driving under broadband conditions.

The stage for force emission (Fe) from the piezoelectric transducer can be modeled by the Mason or KLM equivalent circuits, including real mechanical losses. In fact, the mechanical losses into the piezoelectric stage of the transducer (XT, in Figure 6), emitting a force Fe, are considered [R_losses_ (ω^2^)] in their quadratic character [13,18,19,20,21,22,23]. 

For the modeling of the low-voltage pulsed circuits, used here to assure a secure control (with the needed high currents) of the very fast repetitive switching of the MOS–FET transistor MFT, a specific block described in [9,12,13] was included, fitted to our laboratory HV capacitive driver, employed in the experiments shown in this paper.

Another simpler option that we have applied to non-linear modeling and simulation of tuned capacitive generators of pulsed high-power spikes, is shown in Figure 7.

In this circuital modeling of Figure 7, we include a P-Spice implementation of both the emission and also reception stages, using: (i) a rather simple implementation of the HV spike generator connected to the E/R basic piezoelectric sections of the transceiver, and (ii) the circuital concretion of the emitted and receiver impedance matching layers to the propagation medium; this is taken into account by using transmission lines (T layer 1, 2 in Figure 7b). Here, Rf represents the acoustic impedance of the propagation media, and Rb1 and Rb2 are the acoustic impedances of the E/R transducer backing sections [18,19,20,21]

This other optional scheme is used here to obtain the pulsed current and power in order to drive a 1 MHz transducer for emitting pulsed mechanical waveforms in the time domain, for a case with moderately high voltage. Its driver setting was chosen here in order to illustrate the abovementioned pulsed effects of high-power character. These brief phenomena will be shown in some components and waveforms involved in capacitive-discharge driving stages included in efficient NDE and ultrasonic imaging applications.

## 3. Results for Spikes Generated from our Capacitive HV Drivers and for High-SNR Experimental Ultrasonic E/R Pulses Measured in Wideband T–T Schemes

In order to demonstrate explicit results for specific settings of our pulsed high-power capacitive-discharge drivers, some selected pulsed results will be shown and analyzed in this section, which are indicative of the resulting high SNR. Quantitative ultrasonic waveforms, received after high-power spike driving, will be detailed. These results, selected from efficient driving cases, can be classified as of moderate high-power values. 

Our improvements reached in the main parameters, and also some motional influence on driving spikes, will be shown. For achieving accurate numerical time simulations of these rather complex non-linear high-power driving processes, we implemented precise electrical and mathematical models in time and Laplace domains (described in Section 2.3: Figure 7, Equations (1)–(3)) for an intermediate frequency range. In this way, a precise quantization of the resulting E/R waveforms was obtained, in order to be shown here.

### 3.1. Received Ultrasonic Signals Reachable with Our Tuned HV Capacitive Drivers

Firstly, certain motional influences in an HV driving spike coming from the driven piezoelectric transducer are clearly depicted in Section 3.2, using our experimental and calculation procedures for obtaining the output waveforms from our lab HV drivers. 

The option proposed in this paper, for HV capacitive drivers of wideband transducers, has a very important influence on the received ultrasonic signals in E/R schemes. As a resume of its global efficiency, our measured results of ultrasonic signals (received in T–T mode) are shown in Section 3.3, which reach amplitudes as high as: (i) 76 V_pp_ through a 2 cm-thick methacrylate piece placed between two wideband transducers; (ii) 205 V_pp_ when the transducer faces are put in direct contact. These high values were taken across receiver transducer terminals, without applying any reception amplifier. The capacitive driving parameters (according to Figure 4) were: V_0_ = 310 V, R_D_ = 100 Ω, C = 10 nF, and R_L_ = 1 Ω. 

As values of reference, pulsed echo-signals received in similar ultrasonic applications (usually reported) move in the range of tens/hundreds of Milli-Volts. 

Our reached results suppose a huge SNR increase in the final ultrasonic signals at the reception stage (∆ ≅ 37–46 dB) with respect to usual values. This is due to the efficient electronic capacitive driving, analyzed here. An adequate inductive tuning of transducers also influenced these results by doubling their amplitude and bandwidth, simultaneously [9]. In the above examples, the backed five Mrayl transducers were TecalSA Mod-Q269-270, with lead zirconate titanate ceramic Pz27 (*) and a protective layer, all in a steel case [18].

(*) [Clamped capacitance C_0_^S^: 1.28 nF; charact. impedance Z_0_: 3.79 × 10^7^ Mrayls; long. velocity: Vt: 4618 m/s; piezoelectric constant h33: 1.97 × 10^9^ V/m; and mech. quality factor Qm: 80].

### 3.2. Motional Influences in an HV Driving Spike from the Piezoelectric Transducer

Using a short *t_on_* time in the HV MFT transistor (Figure 4 and Figure 6), some influences can be observed in HV spikes (used for driving), from mechanical (motional) vibrations in a Tecal Mod-Q wideband transducer (1 MHz), which rise as a response to our efficient electrical driving. This curious non-linear motional “feedback” is due to the fast inverse piezoelectric response of the driven transducer (from the internal PZT element) to the high-voltage negative edge, created just after spike launching. In Figure 8 and Figure 9a, this effect can be clearly appreciated, experimentally measured, and calculated on an HV spike.

Figure 9a shows this same driving spike, but now calculated from the model of Section 2.3. The abovementioned non-linear mechanical influences in the spike’s initial microseconds (which are coming from vibrations of the driven transducer) also clearly appear here.

This spike oscillation could be suppressed with a longer *t_on_* in MFT switching. However, due to its short duration and time overlapping with transducer vibrations, they have no influence on the posterior ultrasonic signals related to this spike electrical excitation.

An excellent agreement between the HV spikes of Figure 8 and Figure 9a can be appreciated. These non-linear oscillations have a shape similar to those of the pulse of force emitted by the transducer, whose calculated waveform is shown in Figure 9b. In fact, these electrically induced oscillations during the first 4 µs of the time waveform in Figure 9a are in a close concordance with the mechanical force emitted by the transducer face (Figure 9b).

### 3.3. Measured and Simulated Pulses with Dynamic Ranges Improved by Capacitive Driving

A priori, it is not easy to decide the more favorable values for coupling various driver electrical components with each transducer, in imaging and NDE emission stages, e.g., damping resistance R_D_; emission inductive tuning L_0_; and discharge capacitor C. In this subsection, our useful modeling tools (described in Section 2.3) are applied to that objective.

These can be also used to assist in deciding the optimum efficient driver setting for other transducers and similar testing. Results are shown here related to (i) the pulsed electrical output intensity flowing through an emitter transducer, and (ii) the pulsed power delivered from our tuned HV capacitive driver. This unit has a capacitive circuital scheme as those in Figure 2, Figure 3 and Figure 4, which is needed in order to achieve high dynamic ranges in very efficient NDE and ultrasonic imaging applications.

These first results were obtained by applying our non-analytical modeling, which is a necessary solution due to certain involved circuits showing non-linear behaviors. These conducts are modeled by means of our explicit circuital model including mechanical losses in the piezoelectric stages (Figure 7b), which is valid in this moderate frequency range.

The practical solution for E/R ultrasonic analysis, used here to study the mentioned non-(linear/ideal) responses, includes a global circuital modeling of the all the involved electronic and piezoelectric blocks. In this line of analysis, implementations in the P-Spice context are employed, from two coupled equivalent circuits (Figure 7) with parameters adjusted to each practical case to be analyzed.

Our applicative objective was obtained by numerical calculations of the main waveforms at the output of our capacitive driver (shown in Figure 4) in real loading conditions, and of the subsequent mechanical pulses radiated by the driven transducer, and also by those experimentally received from the propagation medium in resistance, Rf (see Figure 7b). 

An author’s software for analyzing the time domain responses of these pulsed high-power capacitive drivers (electrically matched to wideband transducers), was specifically adapted to our laboratory experiments. In this way, numerical simulations [14,18] were made here of the overall tuned driver and transducer subsystems, involved in our applications for very efficient ultrasonic emission.

At this point, to perform a first analysis of waveforms in the HV pulser output and in the emitting transducer face was the option chosen here, which is quite adequate for estimating the whole behavior of a wideband emitter valid for very efficient pulsed ultrasonic radiations. The emitter piezoelectric transducer selected here had a wide frequency band centered around 1 MHz, and was inductively tuned (L_0_) and electrically damped (R_D_) in parallel with its electric terminals (as shown in Figure 4b).

We calculate the waveforms related to the driving current and the electrical power supplied by our HV capacitive-discharge driver, and also those for the force and velocity radiated from the transducer face, specifically for a typical NDE ultrasonic channel.

The emitter transducer used in these calculations, Mod-Q269 of TecalSA (Madrid, Spain), has a frequency band in its electrical response of rather moderate width, as it can be seen in the measured frequency behavior of both components, conductance G and susceptance B, of its input electrical admittance (Figure 10a).

The HV driving parameters we deduced here in order to obtain the received ultrasonic band, as depicted in Figure 10b, were: V_0_ = 310 V, R_D_ = 100 Ω, C = 10 nF, and R_L_ = 1 Ω.

In relation to the ultrasonic response in through-transmission (T–T) mode joining this Q269 transducer with a very similar device (Q270), through a piece of methacrylate (PMMA), the measured E/R frequency band is shown in Figure 10b. This finally received untuned electrical signal has a rather broad band in accordance with the G conductance band (shown in Figure 10a) of both transducers. 

These two T–T coupled piezoelectric transducers were constructed from lead zirconate titanate Pz-27 piezo-ceramics backed with Rb (1, 2) = 5 Mrayl, and protected with frontal outer layers composed of epoxy and alumina with a thickness close to λ/4 [11,18]. The other transducer data were already detailed at the final part of Section 3, part 1.

One of the broadband transducers, used to compose the ultrasonic received band in Figure 10b (Mod-Q269), was inductively tuned and driven by our HV capacitive driver, in order to obtain intense and rather brief waveforms in its terminals and emission face, as shown in Figure 11. This figure details the calculated waveforms for the pulsed current (Figure 11a) and power (Figure 11b) driving the transducer Mod-Q269 from our capacitive-discharge driver, with the following settings: V_0_ = 395 V, C = 9.3 nF, R_D_ = 470 Ω, R_L_ = 1 Ω, and L_0_ = 56 µH. The resulting emitted pulsed force (Figure 11c) and velocity (Figure 11d) in the transducer emission face are also shown.

The main part of the pulsed waveforms in Figure 11a,b, for time evolution of the electrical current and power delivered to the wideband transducer, have (as desired) a duration near a half-cycle of the series resonance frequency (1.02 MHz), corresponding to the maximum point of the spectral behavior of the transducer electrical conductance [9].

As a result of this efficient electrical driving obtained, our final E/R ultrasonic signals have a huge increase in amplitude and bandwidth, as it is shown in Figure 12a,b.

These two parameters depend not only on the transducer band and electrical driving setting, but also (and very notably) on other emission elements that we introduced for its inductive tuning and for fitting the spike time duration around a half-cycle of the transducer working frequency [9]. These relevant relationships, and their direct consequences on the final frequency bandwidth and resulting dynamic range (of the signals used for ultrasonic imaging and NDE), will be considered in the final part of this section.

The pulsed power supplied to a transducer by our HV capacitive pulser (along thousands of repetitive drivings), with the parametric setting used in Figure 11, shows levels typical of CW high-power ultrasonic applications. These results confirm the principal paper starting point, related to the possibility of achieving very intense values in the pulsed electrical driving and consequently in emitted pulsed forces from driven transducers.

In the following, it is explained how this efficient performance in our emission stage dramatically improves the resulting frequency bandwidth and amplitude measured in the ultrasonic reception stage (Q270), which can be verified in Figure 12a,b. This figure details improvements obtained in received signal waveforms as a result of applying our tuned HV capacitive driver to Q269 transducer into a T–T ultrasonic configuration [9,11]. 

It must be noted that: (i) *the final*
*bandwidth* (Figure 12a) *doubled its initial value* in Figure 10b; and (ii) the amplitude of the rather short ultrasonic pulse received through a plastic piece (made with PMMA) reached the elevated value of 76 V_pp_ (Figure 12b).

### 3.4. High Dynamic Ranges in Aeronautic Industrial NDE Achievable by Capacitive Driving

The potential large dynamic range suggested in the laboratory for our capacitive driver allowed us to resolve a real, very difficult industrial inspection problem, which was pending a solution in the aeronautical sector. We solve this particular NDE problem for two conventional transducers similar to the Q269 probe used in Figure 12, of the same maker (Tecal—L1M20/N1M20), coupled with the piece to be inspected through two water jets. The final aim was to achieve the detection of small adhesion defects in landing flaps of Boeing-777 plane wings [10,17]. The greatest initial difficulty to apply this industrial inspection to wings was derived from the fact that the testing ultrasonic pulses had to pass through two thin external laminates of CFRP composite (resonant at ≈1 MHz) and a very thick zone (140 mm) of non-metallic (*nomex*) inner cores, as can be appreciated in Figure 13. 

This combination of three propagation paths had special problems because it introduces: (a) huge ultrasonic attenuations (a part, through air), which strongly increase with the value of testing frequency employed, and (b) distortions on the received signals by reverberations of testing ultrasonic pulses, aggravated in this type of piece because laminate resonances appear, precisely at a rather moderate frequency range around 1 MHz (value imposed by the airplane manufacturing company, in order to limit the attenuation).

In addition, other quite-high losses on the testing ultrasonic pulses appear, due to some unfavorable industrial conditions in these NDE inspections (e.g., water jet coupling and high EM noise). The set of all these limitations had previously prevented the achievement of efficient NDE in the thicker areas of landing flaps using conventional technology.

Our E/R ultrasonic system for NDE of airplane wings was installed in plants of the EADS-Airbus company, to obtain sharp images of artificial defects in landing flaps with very high inner ultrasonic attenuations. These basic intrinsic attenuations, measured through the thicker zone (in areas without flaws), are near to 95–100 dB for 1 MHz testing, a huge range to which other typical losses in the robotic industrial NDE must be added.

In order to have a margin wide enough to discriminate possible internal defects, *an enormous*
*dynamic range near ≈ 150 dB had to be assured*, and so to be capable of reaching an efficient industrial inspection for quality control of this type of aeronautic piece (see Figure 14).

As a value indicative of the very high amplitude signals, received by a T–T direct contact between transducers L and N 1M20, *pulse voltages as high as 205 V_pp_ could be measured* [10]. In a real NDE case, by applying our tuned capacitive transceiver, the sharp image of Figure 14b was achieved. In its right part (in red), four detected flaws are shown, which were artificially induced by Boeing/McDonnell Douglas in this flap portion.

We previously verified how the global transfer function (of wideband PZT transducers in T–T mode) can be improved in the amplitude and smoothing of band ripples [10]. For this, we replaced a very short spike of rather uniform frequency band (DV(ω) ≈ 1, as a Dirac’s δ) with a Laplace function capable of correcting both aspects. 

From this study, for V_0_ = 365 V and R_D_ = 100 Ω, that function can be roughly expressed as:DV (s) = −340.4 [11.3 × 10^−9^ s^2^ + 1.2 s + 21 × 10^11^ s^−1^ + 1.8 × 10^6^]^−1^(5)

Expression (5) was obtained from our Formulations (2)–(4) described in Section 2.3 to analytically model the inductive tuned capacitive driver designed in our group, SSTU-CSIC.

This specific driving pulse, where “s” is the Laplace complex variable, reached the expected result in a scheme with transducers, such as those in Figure 14a, where we were looking for correcting initial roughness in the T–T compound spectral response by obtaining a band upper part as flat as possible (see Figure 12a), with ≅a 50% of BW at −3dB. 

In order to confirm these excellent results of our HV tuned capacitive generators with other commercial transducers, we show in Figure 15 more laboratory results obtained with two typical commercial probes, used regularly in industrial inspection equipment for automatic testing in order to obtain NDE and imaging: MI0066 and L29160 devices (1 MHz) of KBA Delta Krautkrämer series, which are indicated by their manufacturer for water coupling, and with the emitter already tuned inside its housing. This pair of emitter/receiver transducers were faced coaxially by direct contact with a coupling gel, and the reception conditions were kept similar to previous experiments (electrical tuning of PZT receiver in parallel with the input impedance of an oscilloscope (10 MΩ, 15 pF)).

The emitter transducer was driven with a 300 V untuned spike (10 ns fall time), having an output energy that far exceeds, for example, the maximum delivered by Olympus (Houston, TX, USA) Panametrics 5052UA P/R (a “standard” in efficient laboratory tests). 

The signal received in T–T mode with the Krautkrämer probes is shown in Figure 15a), already with a somewhat improved amplitude (5 V _pp_) with respect to usual NDE results, but with a band “centered” away from the nominal probe frequency (15% below, at least).

Looking for a better configuration of our transceiver for these commercial transducers, after testing various solutions from electrical experimental data, the tuned received amplitude could be notably improved, up to 16 V_pp_ (Figure 15b). Similar values were obtained for two Panametrics probes A314. These amplitudes had been reduced in part by the notable probes backing and the λ/4 layers coupled to water (not to a plastic).

To further increase the efficiency of this emitter–receiver process and also centering its frequency band to the nominal value, we tested other devices of similar frequency (Tecal L&N—1 M-20) with smaller backing and coupled to plastic. The reception was tuned and the driving spike was adapted to the emitter probe. The ultrasonic result is shown in Figure 15c: a working frequency very close to the nominal one of both probes, and a strong increase of 28 dB from Figure 15a, up to 130 V_pp_. These results validate the efficiency of our laborious modeling process by successive approximations, and confirm the utility of the tuned HV capacitive driver analyzed here with different commercial transducers.

In the following and last section of this paper, the above shown graphic results are discussed, in relation mainly to *the*
*method*
*of*
*achieving optimized*
*received signals with very high net dynamic range available (NDRA*), by means of the design of tuned driving spikes delivering pulsed power of some k-Watts to the transducers. The resulting protocol is a very *useful*
*guide*
*for achieving a very high sensitivity in ultrasonic NDE and imaging applications*, even in the more difficult cases, with transducers having medium and low input electrical impedance (e.g., for PZT). Some additional details on the modeling and simulation methods proposed here for precise calculations will also be given in Section 4.

The main advantages of our tuned HV capacitive discharge drivers will be analyzed and quantized, upon other circuital topologies and HV driver types without capacitive storage of energy, which are uniquely used for exciting other types of transducers in some very low-ultrasonic-power applications, which only need a very reduced NDRA, mainly due to high input electrical resistance, i.e., they do not require electrical currents as high as those needed in the PZT probes investigated in this paper.

## 4. Discussion and Conclusions

The waveforms in Figure 11a,b show the electrical results obtained during the capacitive driving (≈0.5 μs) of a wideband piezoelectric transducer, by using our HV capacitive-discharge driver, with a rather moderate spike voltage (400 V) but having other pulsed performances as high as 1600 Watts_pp_ in power and 6 Ampere_pp_ in current. 

This constitutes a useful example of the short-time high-power performances needed in some very efficient applications for ultrasonic imaging and industrial NDE, which could be classified in principle as working in the low-power range. As a result, the establishment of a pulsed high-power repetitive regime for several thousands of shots/s, but with a finally low averaged-energy consumption, has been confirmed for some broadband transducers in the ultrasonic range of the MHz, as was mentioned in the introduction. 

Precise calculated time waveforms were obtained using a careful modeling of the HV capacitive driver designed in the group SSTU-CSIC, with adjustable parameter settings, for very efficient ultrasonic emission processes. This is based on finding the best HV driver parametric settings to generate very intense spikes, from a specific modeling of each practical case. This driver is proposed here for repetitive transducer excitations, based on successive HV capacitive discharges (at a PRF ≥ 5 kHz) over an inductively tuned PZT load connected across its output terminals, which contains the emitter transduction subsystem.

The results in Figure 11a,b are related to the global simulation of the emission process, by combining our circuital models described in Section 2.3 for the driver and for the transducers (Figure 7). Moreover, in the first part of that section, our mathematical approximation is presented, which is based on an analytical expression in the Laplace domain for the HV driving spikes. From this expression, a time response can be obtained by applying the inverse Laplace transform. 

The driving waveforms resulting from these two consecutive calculations will be valid for many ultrasonic applications, except (uniquely) when some inductively tuned schemes are used. In this last case, the modeled driving waveform will only reproduce with accuracy the main part of the driving pulse, i.e., the initial large negative spike. 

However, in these cases, a circuital modeling can be used, with equivalent circuits proposed by us for the HV driver and tuned emitter transducer (Figure 6 and Figure 7). The complete time evolution of the driver output parameters (and their associated waveforms) can be properly calculated and depicted, around the series resonant frequency of the backed broadband transducer coupled to the radiated media by impedance matching layers.

It was also confirmed in this paper, that even though the pulsed power calculated across transducer terminals ranges around k-Watts, the consumed “integrated” power is of only a few Watts. This very intense pulsed regime, during the necessary electrical driving shots in these applications, was performed for the elevated PRF typically used in ultrasonic imaging, and also for other additional NDE experiments working at the bigger PRF of 10 kHz. As additional electrical data referred to the driver output parameters, the pulsed electrical current flowing through the emission transducer can reach up to 10 A_pp_.

The application of non-linear modeling [12,18] and associated simulation software [6,14], both developed by the authors to obtain the electrical parameter values in HV driving outputs and the pulsed mechanical waveforms emitted by the transducer Q269, originates the four pulses depicted in Figure 11. These results prove that our computational tools are adequate to accurately represent the effects of very intense electrical excitations. In fact, the subsequent improved emitted waveforms from the transducer face (force and vibration velocity) were also calculated for an efficient driving (including non-ideal effects). All these simulated waveforms results are coherent with previous reported data. 

Despite the high intensities and non-linearities related to the semiconductors (MOS–FET transistor and several signal diodes, included in the output electronic circuits), our devices have not shown serious heating problems, under this repetitive temporary high-power driving. This is owing to the fact that even though the instantaneous current passing through them is very elevated, during hundreds of nanoseconds (along thousands of times/s), the total average powers dissipated in these electronic components are really quite small. In addition, non-linear risks in the received pulses are rather small, since in detection and imaging applications a harmonic performance remains in the pulsed E/R ultrasonic responses, due to certain filtering effects from both limited-band transducers. 

Saturation effects in the receiver transducer by over-driving were not observed. In order to connect the received signals to linear/logarithmic amplifiers, if this is necessary in particular cases, a voltage divider by 10 or 100 could be included before the amplifier, which approximately will hold the high NDRA measured across transducer terminals. 

These improved pulsed high-power drivers, to be used in high-SNR ultrasonic NDE and HR imaging of inner parts in inspected pieces, must be carefully adjusted to achieve an adequate driving of each emitter PZT transducer, with brief pulses having enough power to achieve high net dynamic ranges in the resulting signals in reception. This high-power is required due to the low working input electrical resistance (tens/hundreds of Ω) of the required efficient transducers in the MHz range. This resistance can be measured at transducer terminals, such as R_motional_ parameter in its mechanical branch, when the transducer radiates in its series resonance (i.e., at frequency of its maximum electrical conductance). 

In order to satisfy the essential power requirement for assuring an efficient PZT emission, driving from a pulsed high-power capacitive generator is needed, such as those that have been analyzed here. This provides short output pulses with high electrical currents (up to 5–10 A) through inductively tuned transducers in the usual working conditions, i.e., mechanically loaded with specific acoustic impedances of several MRayls.

Other reported HV pulse generators of up to 1000 V, designed to drive air-coupled [24] or purely capacitive transducers [25,26], are far from being able to electrically deliver such high-power spikes as those required for the high-efficiency NDE and imaging systems considered here. These pulsed power levels of up to 5000 W_pp_ during hundreds of ns would also be needed for new HR sensing ultrasonic systems to be developed in the future for medical echography and industrial NDE, with very high SNR and dynamic range NDRA. 

The reason for this lastly mentioned requisite is that their transducer emitting faces will be loaded (in the more usual applications) with solid or liquid media, during the hundreds of nanoseconds needed in each HV driving, which will create rather high values in the input electrical conductance during the transducer working periods under series resonance conditions, in order to attain high energetic efficiency in its ultrasonic emission. 

In fact, in the abovementioned papers for capacitive loads, no data are given about either: (i) real pulsed power flowing through the transducer “motional” resistance during its whole driving time, and (ii) the resulting pulsed voltages of measured or calculated ultrasonic signals in the reception stage. Nevertheless, these data are just those needed to confirm high efficiency in any HV driver of wideband transducers.

On the contrary, the results obtained in this paper, about the real delivered pulsed power and the resulting voltages in the received ultrasonic signals, confirm the “special adequacy of our (pulsed high-power) HV capacitive-discharges drivers” to the very demanding power levels required in the efficient wideband piezoelectric applications, such as those that have been researched here. The main paper contributions are: very high (a) signal-to-noise ratios (SNR) and (b) related net dynamic ranges available (NDRA), achieved in both ultrasonic E/R sub-processes. All these aspects are clearly specified and quantized in the following paragraphs.

**Main Data and Conclusions achieved** in our work:▪By a specific setting of our laboratory-programable HV tuned capacitive driver, using the modeling strategy that we propose here, and for a rather moderate spike voltage (V_0_ = 400 V), *the pulsed high-power* passing through the firstly selected wideband transducer, with a λ/4 coupling layer to a PMMA piece, was of 1600 W_pp_ during ≈500 ns, but it *could arrive up to 5000 W*_pp_ with V_0_ = 700 V.▪The *force emitted* by this transducer *for V_0_ = 400 V attains 240 Newton*
_pp_ in its ultrasonic radiating face. This is shown here during the first cycle of the emitted short pulse of force (Figure 11c). From this notable force level promoted by our capacitive driving option, it is possible to easily radiate acoustic fields (along the ultrasonic aperture projection), *with average pressure values* in the order of the *MPascal*_pp_, by *using a simple unfocused transducer.*▪The driving of several wideband transducers employed in this paper with calculated waveforms for high-power spikes (with V_0_ = 395 V), from our tunable HV capacitive driver, made it possible to measure *pulsed*
*ultrasonic waveforms* (in E/R) *having* short time duration and *very high amplitudes*, e.g., 76 V_pp_ at the receiver transducer, in T–T regime through a PMMA piece *(205 V*_pp_, by direct contact). This suggests, in the finally received ultrasonic signals for HR imaging and NDE, *improvements in SNR and NDRA of* up to 38 (45) dB, which could be extended *up to 43 (50) dB* using 700 V spikes (if this voltage was electrically supported by the transducers), *without using any amplification, averaging, or filtering*.▪*These huge voltages* in the received pulsed ultrasonic signals *must be compared with* those acquired in *classical ultrasonic* pulse-echo imaging or T–T NDE cases from inside solid or liquid inspected media, which are usually *below values of 1 V*_pp_
*or even of 100 mV*_pp_ [26,27,28,29,30,31].▪Therefore, from our highly efficient driver (demonstrated by their excellent results) it can be predicted that its circuital topology (deeply analyzed in this paper) will originate pulsed high-power driving spikes very adequate for a *hugely efficient driving of wideband PZT transducers having very low motional input resistance*, through a low series output impedance (≈1–2 Ω).▪Other final aspects to be considered-As a clear practical result: By using our tuned capacitive high-power driver, it is possible to improve the resulting SNR in wideband applications, with an *important*
*increase in the net dynamic ranges available (NDRA)* in ultrasonic imaging and NDE applications, of *up to 70–140 dB,* when *nowadays* these *range*
*≅ between 20 and 70 dB*. The total power needed to achieve such a large NDRA for usual PRF is small: less than 4 W/channel.-*This drastic NDRA improvement can be decisive to make possible ultrasonic inspections with extremely high attenuation or very low acoustic transmission* through some inspected media, such as: lung, foam, nomex, cork, benzene, gases, wood, etc., *where, e.g., losses of at least 75–85 dB used to be suffered*—only through the two short paths—existing between emitting and receiving faces of the matched PZT ceramics and the inspected media. In addition, internal *piezoelectric losses and notable media attenuations should also be added.*-A general consideration: This paper shows that tunable *high-power “capacitive” spike generators driving broadband piezoelectric transducers consume an “averaged” power as low as a few W* (for the thousands of (spikes/s) needed, e.g., in ultrasonic imaging). However, the repetitive pulsed intensities flowing through the transducers can attain more than 6 A_pp_ during all the multiple spike durations, even with rather moderate peak voltages (400 V).-As a very relevant result: The powers delivered from HV capacitive drivers, when designed using our mathematical and circuital models, achieve pulsed levels as high as in high-power ultrasounds. For instance, driving spikes with an effective pulsed power of ≅5 kW_pp_, for a peak voltage of 700 V, can be generated in real loading conditions, at high PRF (≥5000 shots/s). This means a *strong increase of >>40 dB with respect to the*
*usual*
*power values*
*in ultrasonic imaging and NDE.*

*All these results*, drastically improving the current usual HV driver performances, *constitute**a very favorable starting point* to achieve huge dynamic ranges in final ultrasonic signals and images, *in*
*new problematic practical situations for*
*T–T and pulse-echo*
*ultrasonic*
*applications*. Nowadays, some efficient solutions are still pending due to the currently poor sensitivities obtained for cases such as: (a) extremely high attenuations through some propagation media or strong decoupling with them (e.g., lung [32], air/water jet coupling [10,24]); (b) very low sensitivity due to materials constituting the active vibrating part of wideband transducers [27]; or (c) applications in the very high frequency range [31].

*The addition of a supplementary margin >> 40 dB in the final net dynamic range*, by using our tuned HV capacitive driver, is more than enough to achieve an NDRA, making possible the achievement of improved *voltage**levels**(>10,000 higher)* in the received ultrasonic signals. Therefore, *this could ensure (with respect to the “state of the art” in current echography or NDE sensitivity) a sure*
*detection of all the*
*investigated*
*defects under*
*ultrasonic*
*inspection*.

## Figures and Tables

**Figure 1 sensors-21-07178-f001:**
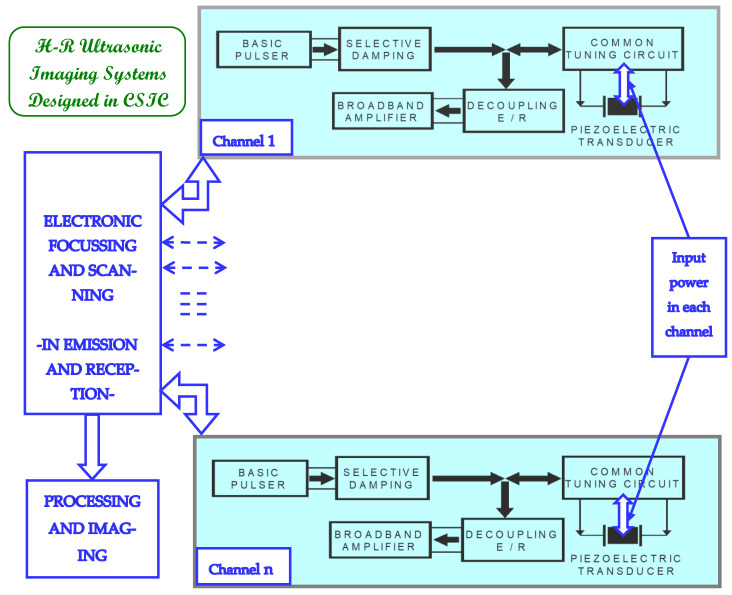
Structure of our HR ultrasonic imaging systems designed in CSIC, which contain a number of controlled E/R ultrasonic channels (from 16 to 128). It usually includes *n* analogic transceivers digitally controlled by complex systems for electronic focusing and HV scanning in emission and reception stages, and a unit for processing and displaying of the received signals and images.

**Figure 2 sensors-21-07178-f002:**
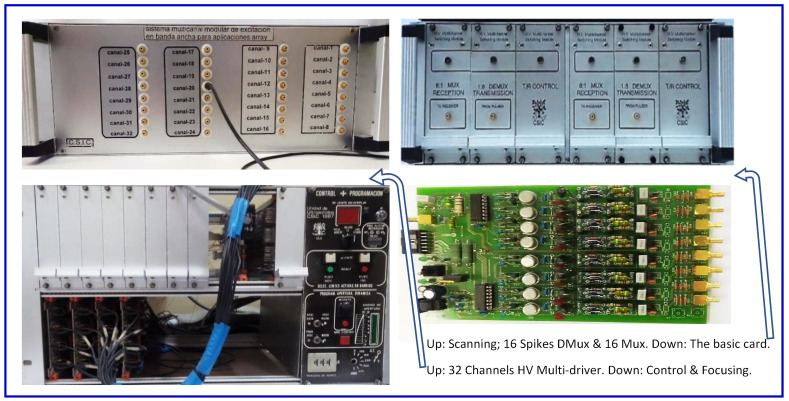
Lab equipment of own design for capacitive multi-driving, electronic focusing, and HV scanning [15] of PZT transducers.

**Figure 3 sensors-21-07178-f003:**
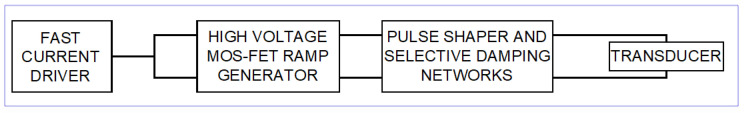
Block diagram of our HV pulsed driver for efficient excitation of broadband transducers.

**Figure 4 sensors-21-07178-f004:**
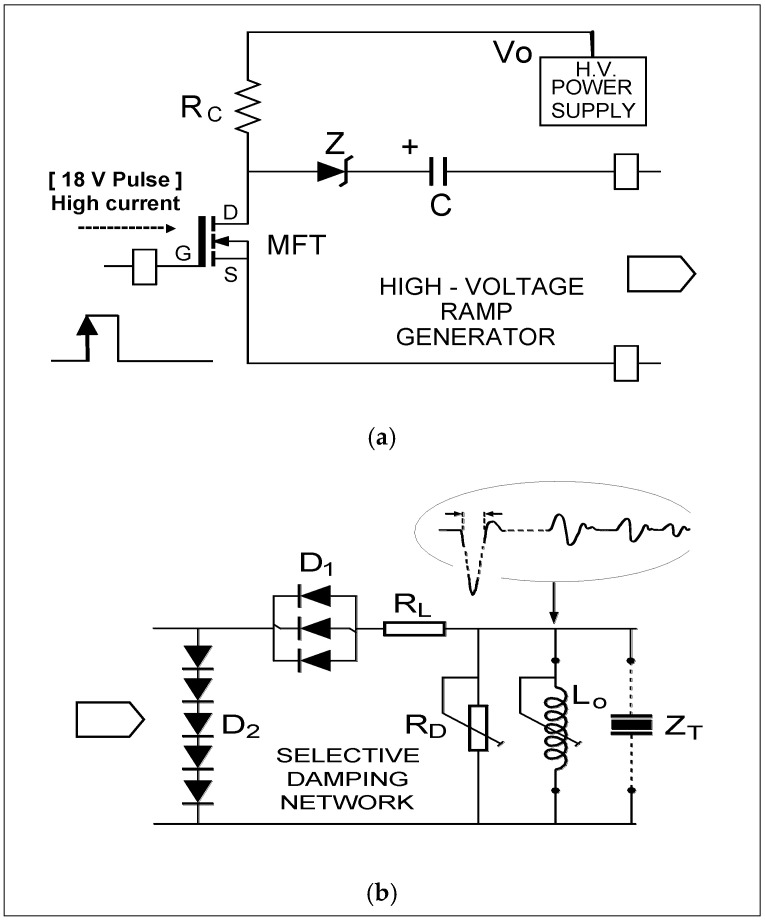
Circuital schemes for the blocks involved in Figure 3: (**a**) capacitive ramp generator and (**b**) pulse shaper and selective damping.

**Figure 5 sensors-21-07178-f005:**
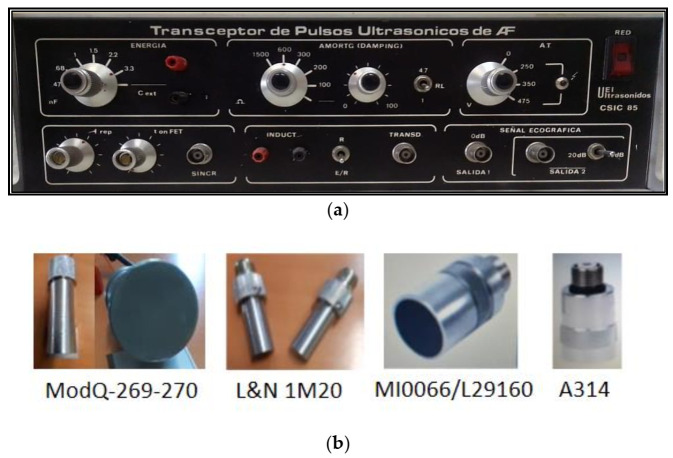
Our own-design reconfigurable HV electronic transceiver (**a**) and wideband piezoelectric transducers (**b**) that have been employed in the experimental works shown and analyzed here.

**Figure 6 sensors-21-07178-f006:**
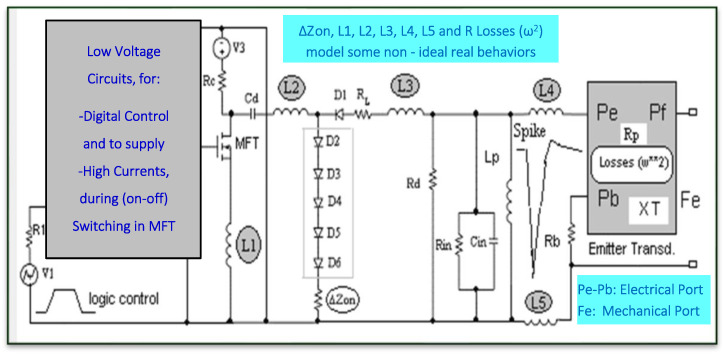
Scheme of our global circuital modeling for emitter responses simulation under high-power capacitive driving in the high-frequency range, with electrical non-ideal behaviors and quadratic piezoelectric losses in XT.

**Figure 7 sensors-21-07178-f007:**
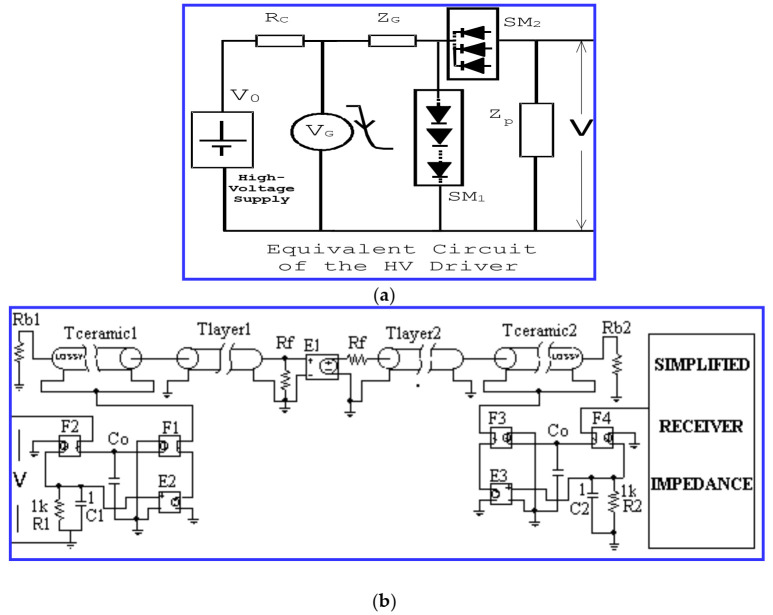
Scheme of a practical option we propose for circuital modeling (in P-Spice) of ultrasonic transceivers under pulsed high-power capacitive driving, for intermediate frequency ranges: (**a**) HV capacitive driver, (**b**) piezoelectric subsystems with mechanical piezoelectric losses in E/R mode.

**Figure 8 sensors-21-07178-f008:**
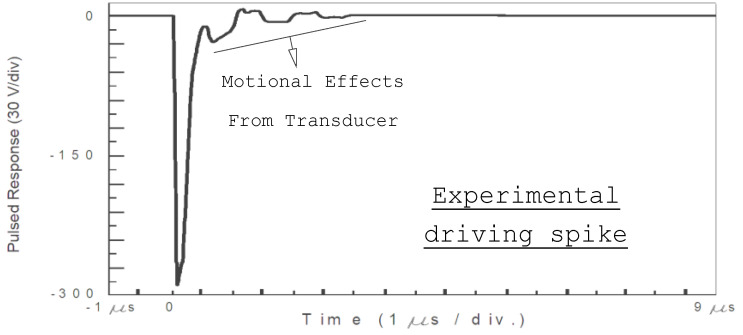
Measured HV spike waveform when driving a broadband piezoelectric transducer in the MHz range (Q269), showing mechanical influences from transducer vibrations during the first 4 µs.

**Figure 9 sensors-21-07178-f009:**
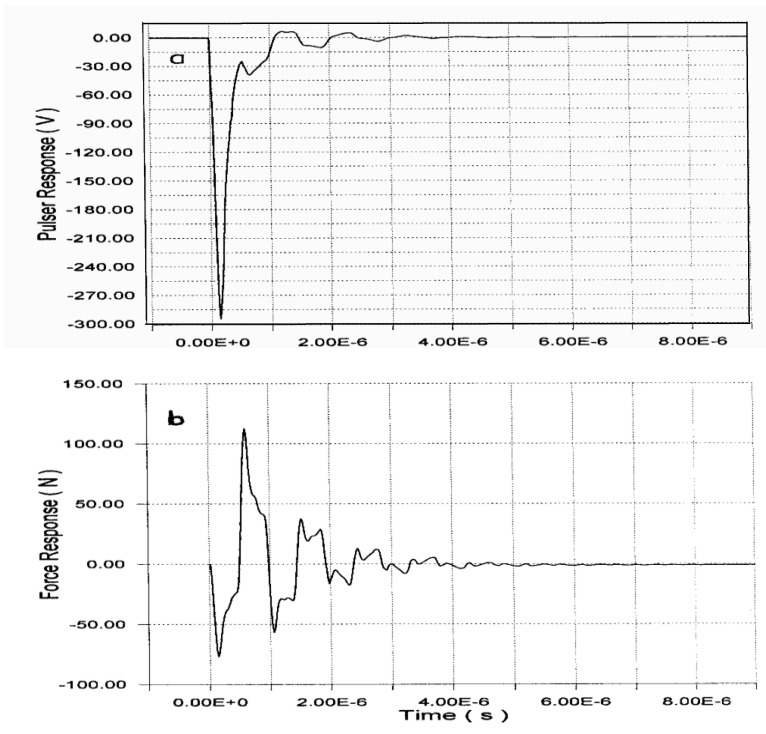
(**a**) Computer simulation of the HV driver spike related to the measured HV waveform of Figure 8, also showing the motional effects from the driven transducer, superimposed on the spike. (**b**) Calculated transient motional force over the transducer radiating face, originating those effects.

**Figure 10 sensors-21-07178-f010:**
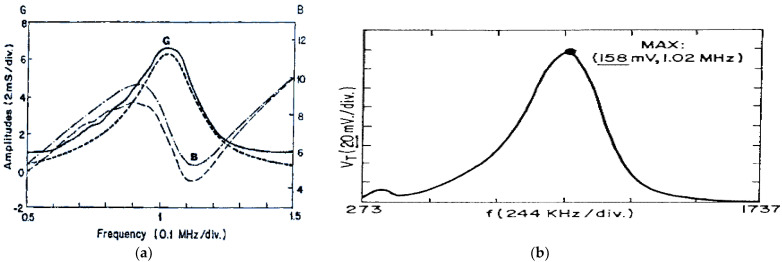
(**a**). Measured frequency spectra of real (G) and imaginary (B) parts of the electrical admittance in the untuned Q269 broadband transducer. (**b**) Experimental basic frequency band of the received signal in a Q270 transducer, connected in a through-transmission scheme, without any inductive tuning in either stage, and when Q269 is driven by our HV capacitive driver.

**Figure 11 sensors-21-07178-f011:**
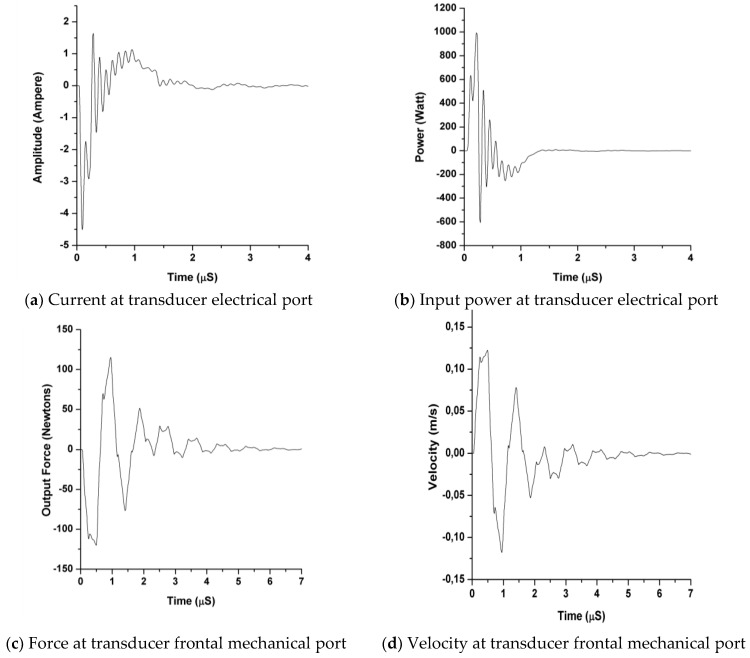
Output current (**a**) and power (**b**) waveforms simulated for our tuned HV capacitive driver loaded with the Q269 transducer, using our circuital models of Figure 7. Resulting force (**c**) and velocity (**d**) pulses, calculated for the transducer face emitting on a methacrylate (PMMA) plastic medium.

**Figure 12 sensors-21-07178-f012:**
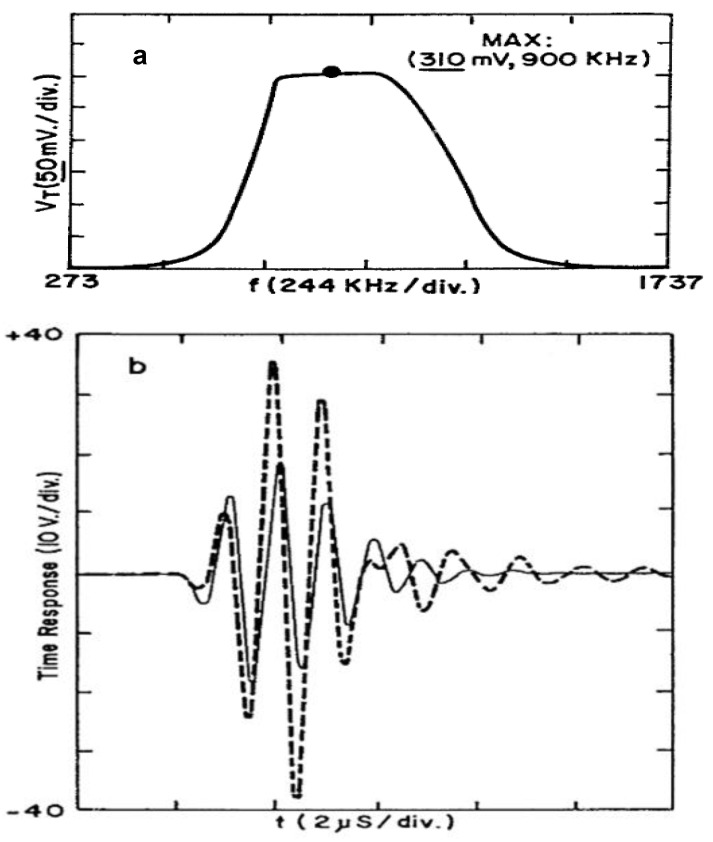
Improvements in the ultrasonic signal measured in T–T mode through a PMMA piece, due to our tuned HV capacitive driver: (**a**) bandwidth; (**b**) time response (in dashed line, with tuning).

**Figure 13 sensors-21-07178-f013:**
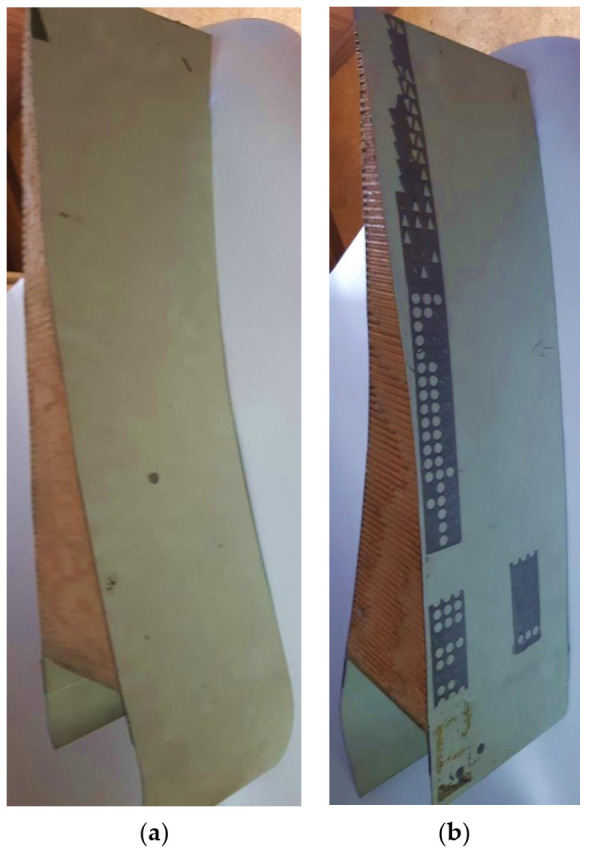
NDE test pieces (landing flap transversal portion, containing mainly air) of (**a**) Boeing-777 plane wings, fabricated for the maker (**b**) with internal artificial flaws located in unknown ubications.

**Figure 14 sensors-21-07178-f014:**
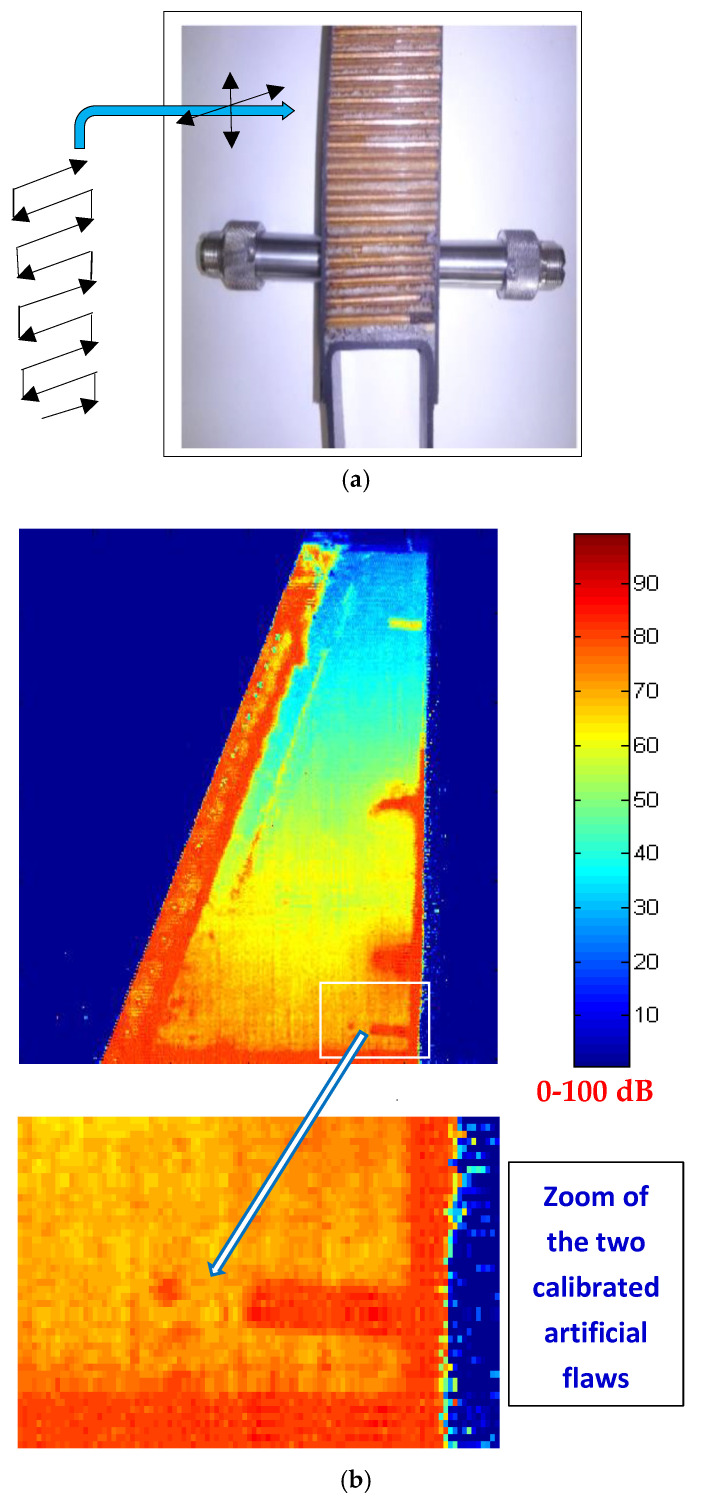
Application of our tuned capacitive driver for NDE of a portion of landing flap in Boeing-777 wing. (**a**) Laboratory-scale mock-up showing how ultrasonic C-Scan imaging was made, with two T–T transducers (here, the real piece of Figure 13 and water jets are not shown). (**b**) The very sharp imaging that was achieved, overpassing losses (into the inspected flap and water jets) of up to 150 dB.

**Figure 15 sensors-21-07178-f015:**
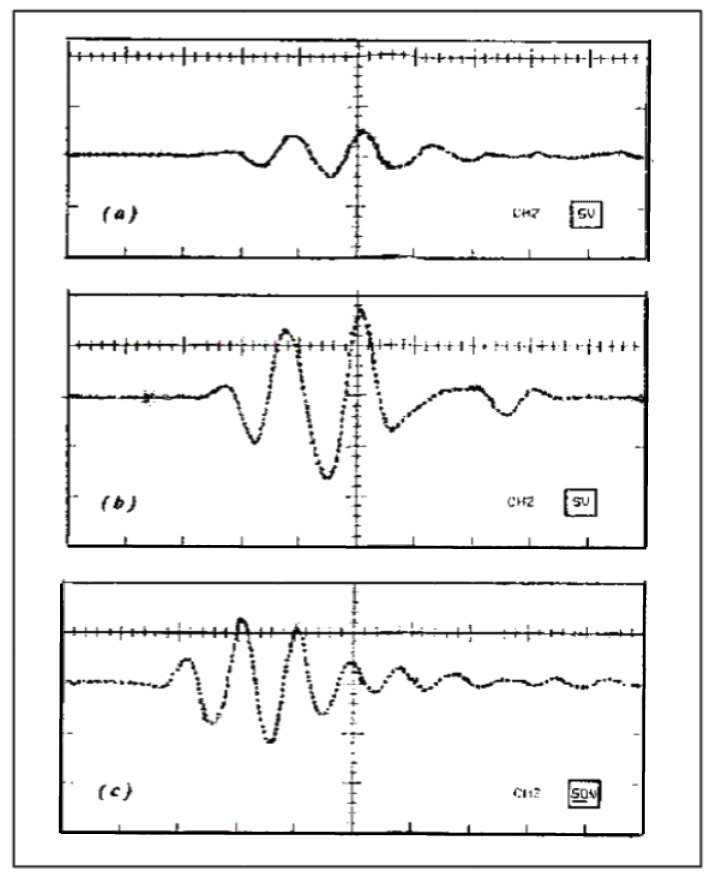
Our T–T received signals with both probes in direct contact: (**a**,**b**) KBA Delta Krautkrämer probes (**5** V/div.). (**c**) L-N Tecal probes (**50** V/div.). The driving was tuned in (**b**,**c**) responses. In all three cases: 1 μs/div. in horizontal axes.

## Data Availability

Not applicable.

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
