# Peer review of "Modeling Pulsed High-Power Spikes in Tunable HV Capacitive Drivers of Piezoelectric Wideband Transducers to Improve Dynamic Range and SNR for Ultrasonic Imaging and NDE"

_sensors, 2021, doi:10.3390/s21217178_

Round 1
Reviewer 1 Report
The authors did a comprehensive analysis of high power driving circuit for ultrasonic transducer especially made in PZT materials. The analogy model such as KLM transmission line is also proper. However, the parameters involved are not listed clearly e.g. PZT properties. Also lack of experimental data is also a major flaw of the research.
The language is not following the style of technical writing and the English needs serious editing. There are many typos or grammar errors can be found easily. The rendering of figures is sloppy, e.g. in Figure 4 the background color and fonts are not proper.
The above problems have to be resolved before the manuscript can be published.
Author Response
Please, see the attachment

Reviewer 2 Report
Dear authors,
your paper intitled "Modeling Pulsed High-Power Spikes to Drive Piezoelectric Broad-band Transducers Improving SNR in Ultrasonic Imaging & NDE" is well organized and interesting, but from my point of view the structure of the transducer is needing. Check for the remarks/suggestions:
Major revisions:
- add a summary of the next sections at the end of the "Introduction" section;
- there is no drawings/structure about the transducer. No references about Matching layer, backing, thermal drain?, array. Please add this important part.
Minor revisions:
- #8: acronym of NDE #8, not in #30;
- Figure 3: yellow strip in the top?;
- #167: replace "microsecond" in the International unit of measure;
- #342: refuse. From "y" to "and"?.
Best regards.
Author Response
Please, see the attachment

Reviewer 3 Report
The paper is interesting and certainly deserves to be considered. Its structure is well established and the methodological rigor with which the problem has been faced is satisfactory.
Some typos are present in the text. Please remove them.
The introduction is too long and complex. I advise authors to break it down into small sections so that readability improves.
Figure 1 is very interesting and certainly deserves special attention. Therefore, I advise the authors to improve the associated caption to make it self-explanatory.
How was (3) obtained? Is it possible to report some specifications?
Author Response
Please, see the attachment

Round 2
Reviewer 1 Report
I did see lots of improvement on the writing style by the authors' effort. There are still some minor mistakes, e.g., @ line 398: "....by emulating the cancelling (to put at “cero”)....".
Another flaw is that still no experimental data can be found. From what I see the circuit are not difficult to built on bench. The authors should do some experiments and verify it on any transducer with known construction and material properties.This is a basic for any engineering research.
Author Response
Please, to see the attachment

Reviewer 2 Report
Dear authors,
your paper intitled "Modeling Pulsed High-Power Spikes to Drive Piezoelectric Broad-band Transducers Improving SNR in Ultrasonic Imaging & NDE" can pass to minor revisions.
Please reduce the section (34 lines are too much) about the summary of the next sections. In 3-4 lines the authors have to realize this (check for the literature).
Best regards.
Author Response
In relation to your comments about the extension of the part: "A summary of the next sections", we have dramatically reduced this paper part.
Best regards.
Antonio